

# Application of collaborative filtering algorithm based on time decay function in music teaching recommendation model

Yina Zhao[1] and Xiang Hua[2]

[1] Yuzhang Normal University, School of Music and Dance, Nanchang, China
[2] School of Education, Dankook University, Yongin, Gyeonggi, Republic of South Korea

## ABSTRACT

To address the issues of data sparsity, scalability, and cold start in the traditional teaching resource recommendation process, this paper presents an enhanced collaborative filtering (CF) recommendation algorithm incorporating a time decay (TD) function. By aligning with the human memory forgetting curve, the TD function is employed as a weighting factor, enabling the calculation of similarity and user preferences constrained by the TD, thus amplifying the weight of user interest over a short period and achieving the integration of short-term and long-term interests. The results indicate that the RMSE of the proposed combined recommendation algorithm (TD-CF) is only 8.95 when the number of recommendations reaches 100, which is significantly lower than the comparison model, which exhibits higher accuracy across different recommended items, effectively utilizing music teaching resources and user characteristics to deliver more precise recommendations.

# INTRODUCTION

With the rapid development of the Internet, big data, and mobile technology, these advancements bring convenience to people while the volume of data continues to grow exponentially. It is increasingly challenging for users to find the resources they need from this vast expanse of information. To address the problem of information overload and assist users in extracting data efficiently, recommendation systems have been proposed (*Tang & Zhang, 2022*). Currently, recommendation systems are a mature and successful technology, deeply integrated into various aspects of Internet products, such as short video platforms, music and movie websites, e-commerce, social platforms, and online e-books. The growing abundance of song resources in music makes it difficult for users to find songs of interest quickly. Additionally, search results must account for users' individuality and varying song preferences, leading to low user satisfaction (*Deldjoo, Schedl & Knees, 2021*).

Amid the rapid rise of educational informatization, online learning platforms are progressively transforming the education landscape, offering distinct advantages over

Corresponding author
Xiang Hua, 15309488003@163.com

traditional teaching methods. These platforms provide new learning opportunities for countless knowledge-seeking students by breaking through the constraints of conventional classroom environments. Integrating personalized recommendation systems into online teaching resource platforms is vital to this transformation. These systems offer customized services by analyzing user preferences and delivering tailored teaching materials to match their interests (*Shi & Yang, 2020*).

However, traditional collaborative filtering (CF) recommendation algorithms have limitations. They primarily rely on users' historical behavior and scoring data to infer interest preferences while ignoring crucial contextual factors like time, location, and mood. These contextual elements significantly influence user behavior and can greatly enhance the accuracy of recommendations. For instance, user preferences in learning and content consumption may shift based on time and environment. Despite these influences, existing recommendation models in educational settings have not sufficiently explored the impact of time-sensitive contextual data (*Qu, 2024*; *Shakirova, 2017*).

Despite their success, existing recommendation algorithms fail to consider dynamic factors such as time adequately. The absence of time-related variables reduces the effectiveness of personalized services in educational platforms, potentially leading to suboptimal user experiences. This research aims to bridge this gap by introducing a time-sensitive dimension to collaborative filtering, improving the alignment of recommendations with users' evolving interests.

In this research, we focus on addressing the gap by integrating time context information into the traditional CF recommendation algorithm. Including time-dependent factors is vital, especially in music recommendation systems, where users' preferences evolve with time and context (*Rabiu et al., 2020*). This concept applies primarily to educational platforms, where students' learning interests fluctuate based on time-sensitive influences such as assignment deadlines or exam periods. Therefore, we propose an improved CF recommendation algorithm based on time decay (TD), known as TD-CF. This algorithm incorporates time context data by fitting it to the forgetting curve of human memory, thus assigning a weight to user preferences based on recency. The key research questions guiding this study are as follows:

(1) How can integrating time context information enhance the accuracy of recommendation systems on educational platforms?

(2) What improvements can be achieved by utilizing a time decay function to model users' short-term and long-term interests?

(3) In what ways can this enhanced recommendation system better serve students with fluctuating learning patterns over time?

## RELATED WORKS

With the development of recommendation technology, it has become a trend for global Internet companies to apply recommendation technology to music recommendations to meet users' personalized music needs. This trend also makes experts and scholars pay more attention to recommendation technology in music.

Personalized recommendation systems are central to enhancing user experiences across various platforms, and integrating contextual factors is a critical step toward optimizing recommendation outcomes. One of the primary challenges in this area is accounting for changes in user interests over time. Contextual environment factors, such as time, location, and user mood, are pivotal in refining recommendations and improving accuracy (*Thanh, Ali & Le, 2017*). A growing body of research has focused on introducing time-sensitive modifications to traditional CF algorithms to capture these evolving interests better. *Ma et al. (2020)*, explored the concept of a scoring time window, where a fixed time frame is used to gauge users' short-term preferences. However, a notable limitation of this approach is the inability to account for variability in users' interest changes. Some users may experience shifts in their preferences over shorter or longer intervals. To address this, *Chang et al. (2021)* proposed a dynamic time window approach that adjusts based on individual user behaviors, offering a more accurate reflection of interest changes. Although this dynamic time window improves flexibility, it still struggles with the challenge of optimally setting the sliding window size for all users.

Other studies have explored scoring time weighting methods, which assign varying importance to items based on their scoring times. *Zhao & Sun (2022)* introduced a time-weighting function, proposing that items rated closer in time receive higher weights, while those rated further apart are given lower weights. This method adds flexibility by differentiating the importance of items, yet it assumes that all user behaviors follow a similar pattern, which may not be universally true. *Chen, Hsu & Lee (2013)* expanded on this by incorporating a recommendation system that adjusts based on users' temporal acceptance abilities, demonstrating its effectiveness in sparse data environments. *Ba et al. (2024)* introduced a scoring-based training model centered on specific target times, while *Vinagre, Jorge & Gama (2015)* also highlighted the potential of time-sensitive data to enhance recommendation quality. *Jiang et al. (2024)* advanced the field by introducing temporal behavior as a factor in collaborative filtering algorithms, applying singular value decomposition (SVD) to improve recommendation precision. Similarly, *Maio et al. (2018)* proposed a CF algorithm based on time-weighted neighbors, where recent data is given a higher weight, leading to improved prediction accuracy. *Liu et al. (2010)* extended these ideas by focusing on the importance of time-sensitive data, assigning greater weight to more recent user behaviors.

Based on the time factor, the latest research further extends the recommender system's ability to capture users' dynamic behavior. *Wang, Wang & Zhao (2023)* proposed a multi-modal recommendation algorithm, which integrated the user's multi-dimensional information, such as time, location, and device, into the model, significantly improving the accuracy of recommendation and user satisfaction. However, the computational efficiency of the proposed method on large-scale datasets still needs to be further optimized. Similarly, *Ko et al. (2022)* discussed the role of user emotion in the recommendation system. They proposed a recommendation model based on emotion analysis, which can capture the impact of user emotion changes on recommendations. Still, its performance in the depth of personalization is not perfect. *Choe, Kang & Jung (2021)* adopted a time series recommendation model based on deep learning. They modeled the user's historical

behavior through the long short-term memory (LSTM) network, effectively improving the recommendation effect in complex situations. However, the performance of this model still has shortcomings in dealing with data sparsity.

While these approaches have advanced the understanding of time-based recommendations, several key limitations remain. First, many models, such as those based on fixed or dynamic time windows, assume that user preferences change uniformly, overlooking the unique and potentially non-linear ways individuals interact with content. Furthermore, the reliance on past behaviors to predict future interests neglects the role of more complex contextual factors, such as emotional states or task urgency, which are challenging to model. The primary focus on time as a context factor has also limited exploration into how other elements, such as location or social context, could influence recommendations in educational platforms. Finally, most algorithms offer limited scalability when applied to large-scale datasets, as they tend to become computationally expensive and struggle with performance in environments characterized by sparse user interactions.

While considerable progress has been made in improving the accuracy of recommendation systems through time-sensitive approaches, the failure to incorporate other dynamic context factors, such as multi-dimensional user behavior or real-time interaction data, continues to pose a challenge. This research addresses these gaps by proposing a comprehensive method integrating time and preference factors to enhance recommendation accuracy in personalized educational platforms.

# DESIGN OF MUSIC TEACHING RECOMMENDATION MODEL

## Overall design

Figure 1 depicts the architecture of the music resource recommendation module, a comprehensive system designed to provide users with personalized music recommendations. The recommendation model processes primary user and resource information and rating data to generate tailored recommendation results. The customized recommendation process is divided into three key steps: data processing, recommendation calculation, and prediction recommendation.

Step 1: Data processing

In the initial stage, referred to as Part A, the system preprocesses data from three primary sources: the user information database, the resource information database, and the scoring information database. This involves cleaning, organizing, and structuring the data to build a robust model. Key outputs of this stage include user feature vectors, resource attribute vectors, and resource attribute preference vectors. These vectors serve as the foundation for calculating user similarity in later stages.

Step 2: Recommendation calculation

Building on the data processed in Part A, Part B calculates user similarity. This is done using the recommendation model, which identifies the nearest neighbors for each user. The model considers user characteristics and an information bias model to predict preferences for new users. The model utilizes rating data and resource attributes to determine preferences for existing users. The aim is to find users with similar tastes and preferences, facilitating more accurate recommendations.

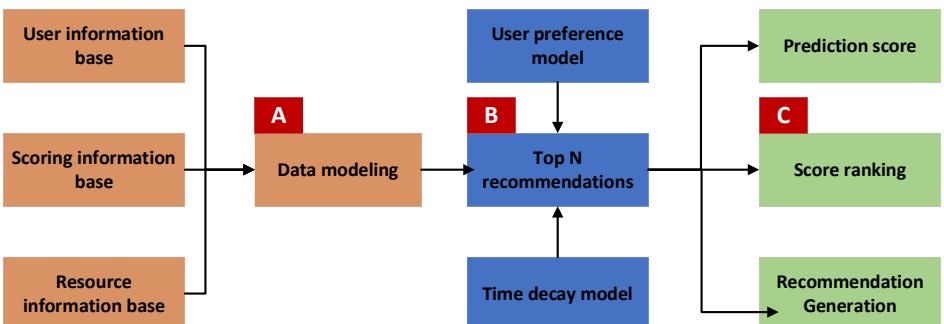

**Figure 1** **Architecture of music teaching recommendation module.**

Step 3: Prediction and recommendation

In the final stage, Part C, the system predicts user scores for various resources. Based on these predicted scores, it compiles a list of each user's top N recommended resources. This step ensures that users receive the most relevant and appealing recommendations, enhancing their overall experience with the platform.

## Recommendation model based on CF
### User-music score matrix

In the CF recommendation model, user preference is represented by a vector. A user-music matrix can be constructed by combining the preference vectors of all users. This matrix is the input of the collaborative filtering model. The goal of the model is to calculate the similarity between users, retrieve the neighbor users of all users, generate a neighbor user set, and finally obtain recommended users and recommended music in the neighbor user set. This "preference" level is expressed as the number of repetitions in a music system with no direct music rating mechanism.

Assuming that the music set that the neighboring users like but the target users haven't heard is M, the implicit scores of all neighboring users on the music in M can be averaged as the predicted score set S of the target users on the music in M. Finally, the music with a TopN score in S can be added to the recommendation list.

One effective method for addressing the cold start problem, particularly for new users, is leveraging implicit feedback from initial interactions. This approach relies on monitoring and analyzing the user's early behavior within the system, such as the music they listen to, the tracks they skip, or the songs they play repeatedly. By collecting and interpreting these implicit signals, the recommendation system can begin to infer user preferences even when explicit ratings or long-term interaction data are unavailable. This method has the advantage of being non-intrusive, as it does not require users to provide direct input about their preferences. Additionally, implicit feedback reflects genuine behavior more, capturing real-time decision-making processes during music consumption. As the user continues to interact with the system, the accumulated implicit feedback can be used to refine recommendations and gradually personalize the user experience.

To effectively implement this strategy, it is essential to define appropriate metrics for implicit feedback, such as the number of times a user plays a song or the duration they listen to specific tracks. The system can build a more accurate profile of user preferences by assigning weights to different types of interactions—such as treating repeated plays as a more robust signal of interest than a single listen. Moreover, advanced machine learning techniques, such as matrix factorization or collaborative filtering, can be applied to the implicit feedback data to recommend music based on patterns learned from other users with similar behavior. Over time, the recommendation system can enhance its ability to predict user preferences, even without extensive historical data.

### User similarity calculation

The cosine similarity calculation can convert the user-music score matrix into a vector in the N-dimensional music space. The score of music that the user has not listened to is set to 0, and the cosine value of the angle between different user vectors is calculated to represent the similarity of users. Because the cosine similarity does not consider the various standards of things among users, the benchmark points of scoring are not the same, so the calculation results have their limitations. Therefore, using the modified cosine similarity. By calculating the difference with the average score of the items evaluated by users, the error caused by the user's rating benchmark point is solved. In this paper, the average score of the song is subtracted from the score of the target music by all users as the new score, and then the cosine similarity is calculated. As shown in Eq. (1), the user rating similarity $\text{Sim}_s$ is:

$$\text{Sim}_s\left(U_i, U_j\right) = \frac{\sum_{s \in I_c}\left(R_{is} - \overline{R_i}\right)\left(R_{js} - \overline{R_j}\right)}{\sqrt{\sum_{s \in I_a}\left(R_{is} - \overline{R_i}\right)^2}\sqrt{\sum_{s \in I_b}\left(R_{js} - \overline{R_j}\right)^2}} \tag{1}$$

where $U_i$ and $U_j$ represent user $i$ and user $j$, respectively, $I_c$ is the intersection set of music that users $i$ and $j$ have heard, $I_a$ and $I_b$ are the set of historical songs of users $i$ and $j$, respectively, $R_{is}$ and $R_{js}$ represent the rating of a song by users $i$ and $j$, respectively, and $\overline{R_i}$ and $\overline{R_j}$ represent the average rating of a song by users $i$ and $j$, respectively.

Calculating the similarity of user tags by using Jekard similarity coefficient $\text{Sim}_t$, as shown in Eq. (2):

$$\text{Sim}_t\left(U_i, U_j\right) = \frac{|T_i \cap T_j|}{|T_i \cup T_j|} \tag{2}$$

Where $T_i$ and $T_j$ represent the set of labels for users $i$ and $j$, respectively, to sum up, the similarity of user attributes S can be obtained by using the two similarity degrees, as shown in Eq. (3).

$$\text{Sim}\left(U_i, U_j\right) = w_s \times \text{Sim}_s + w_t \times \text{Sim}_t. \tag{3}$$

Among them, $w_s$ and $w_t$ represent the weights of similar scores and similar labels, respectively.

By testing various combinations of *ws* and *wt*, the optimal balance between the two similarity measures can be identified, allowing the system to more effectively capture both

the numerical and categorical aspects of user preferences. In some contexts, such as when users provide more categorical data (*e.g.*, genre preferences) than explicit ratings, higher weights may be assigned to the Jaccard similarity measure. Conversely, more emphasis may be placed on cosine similarity in systems with dense rating data.

## Time decay function

Fitting the TD (*Chen, Hui & Thaipisutikul, 2021*; *Huang et al., 2023*; *Liu et al., 2024*) by analogy with the forgetting law of human memory, the TD expresses the weight of users' interest in items, where the longer a user acts on items, the less it will affect users' current interest.

Using the forgetting curve to fit the TD factor provides a more realistic representation of memory and relevance in models. Human memory tends to retain new information more clearly, but without regular reinforcement, even recent information fades over time. By applying this principle to the TD factor, algorithms can prioritize newer or frequently reinforced information while gradually diminishing the influence of outdated data. in recommendation systems, recent user interactions would have a more substantial impact on predictions than interactions that occurred further in the past.

The fitting method of the attenuation function can be linear, exponential, and logarithmic. In this paper, the linear time decay function $f(|T - t|)$ is adopted, and its expression is as follows:

$$f(|T - t|) = \frac{1}{1 + \alpha|T - t|}. \tag{4}$$

Among them, $\alpha$ represents the time decay factor, which decides the attenuation rate of the function. Adjusting the value can simulate the memory curve; if the user interest changes faster, then the decay rate is faster, and the value of $\alpha$ will increase; T indicates the current time, while t is the time of the action; $|T - t|$ represents time difference, and $f(|T - t|)$ decreases as the growth of the time difference, namely value range is (0, 1).

TD-CF needs to increase the weight of users' recent behaviors, and TD is used as a weight factor to constrain the similarity of users. Compared with previous behaviors, users' recent behaviors can better reflect their current interests. Considering the influence of time, the traditional similarity calculation of CF is improved, and the optimized cosine similarity calculation formula is as follows:

$$\text{Tsim}(i, j)_{\cos} = \frac{\sum_{u \in N(i) \cap N(j)} f(|T - t|)}{\sqrt{|N(i)||N(j)|}}. \tag{5}$$

In addition to incorporating the TD into the similarity calculation, the influence of time information on the preference degree should also be considered. Compared with the long-term behavior of users, the recent behavior of users can better reflect the current interests of users. Therefore, when predicting the user's current interest, the weight of the recent feedback items should be increased, and the items similar to those that the user likes or has purchased in the near future should be recommended first. The user's interest preference can be calculated after the time the user acts on the items is obtained. The

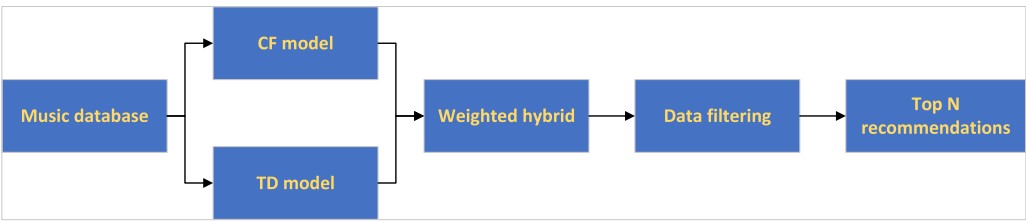

**Figure 2** **Weighted mixed recommendation process.**

modified preference degree is shown in Eq. (6),

$$\text{Tp}(u,i) = \sum_{j \in S(i,N) \cap N(u)} \text{Tsim}(i,j)_{\cos} \frac{r_{uj}}{1+\beta|T-t|} \tag{6}$$

where $\beta$ is the updated time decay factor, which needs to be adjusted according to different data sets.

## Model mixing

The model based on CF is sparse and needs a lot of historical data, while the model based on TD does not entirely depend on the new users' listening history. This paper uses the mixed recommendation model to weigh the two models. When the user's listening history is rich, the weight of the CF model is increased, and when the user's listening history is low, the weight of the time decay model is increased. The flow of weighted recommended music is shown in Fig. 2.

The Recommendation degree rec of the weighted mixed music recommendation model is shown in Eq. (7).

$$\text{Rec} = \alpha \text{Sim} + (1-\alpha)\text{Tp}(u,i) \tag{7}$$

where $\alpha$ is the weight. When the user has a rich listening history, the weight of the collaborative filtering model is increased; when the user has a small listening history, the weight of the time decay model is increased.

# EXPERIMENT AND ANALYSIS

## Data sources

In this experiment, the internationally published music data set "Last.fm Dataset-1k" is used for experimental analysis (https://zenodo.org/records/4589071, doi: 10.5281/zenodo.4589071), which is the representative of implicit feedback data set with context information. By integrating the data of the preprocessed original files into one file, the experimental data set contains 12,293 users, 320 singers, and 18,698 related records. In the experiment, 70% of the data sets are selected for training and 30% for testing. The user interest model is established on the training set, and the method's performance is verified on the test data set.

## Evaluation metrics

The study employs several evaluation metrics to assess the performance of various recommendation models, including root mean square error (RMSE), F1-score ($F1$), and mean average precision (mAP).

RMSE is a widely used metric for measuring the accuracy of predictions in recommendation systems. It calculates the square root of the average of squared differences between predicted and actual ratings. The lower the RMSE value, the more accurate the model's predictions are.

The F1-score is a harmonic mean of precision and recall, providing a single measure that balances both. Precision is the fraction of relevant items among the recommended ones, while recall is the fraction of relevant items recommended out of all possible relevant items. The F1-score is particularly useful in cases of an imbalance between precision and recall.

Mean average precision (mAP) is a metric that evaluates the ranking quality of the recommendation model. It calculates the average precision of recommendations across all users and then takes the mean of these values. Precision is computed at every position in the ranked list of recommended items, considering only relevant items. mAP emphasizes the importance of returning relevant items earlier in the list, providing a more nuanced measure of recommendation quality.

Memory usage (in MB) measures the memory each model consumes. User-Item CF scales poorly with data size, requiring more memory as the dataset grows. Computation time (in seconds) measures the time each model takes to generate recommendations. These measures evaluate the computational efficiency of the models.

## Model comparison

The CF model, TD model, TD-CF model, the singular value decomposition (SVD++) model, and K-means model are compared and analyzed. RMSE is used for evaluation. The comparison results under different sample sizes are shown in Fig. 3:

According to the experimental results, the RMSE value of the prediction result of the mixed model is lower than that of the single model, SVD++ model, and K-means model, which indicates that the recommendation model considering time decay has a better recommendation effect. As the number of predicted scores increases, the RMSE value decreases. From the above analysis results, it can be seen that in the traditional recommendation process, there are still some problems, such as a sparse rating matrix and complex recommendations for new users. Using the improved CF algorithm to recommend teaching resources can solve the problems of data sparsity, scalability, and cold start of users.

The comparison of experimental results in Fig. 4 shows that the evaluation index values of $F1$ and mAP of the proposed model on the competition data set are better than those of other methods. Among them, when the length of the recommendation list is 10, 40, 70, and 100, respectively, the maximum $F1$ value of INCF is 24.59% higher on average than that of the other recommendation models (excluding popular recommendations). The minimum average value is 16.28% higher than the other recommendation models.

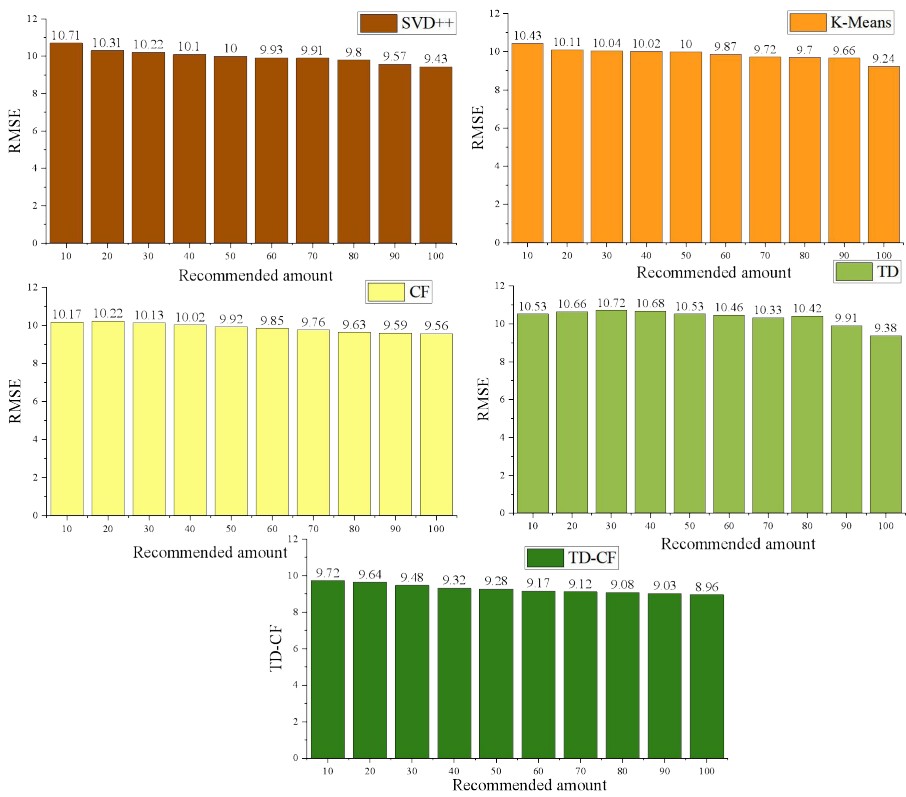

**Figure 3  Comparison of RMSE of different models.**

Mapp is also far ahead, indicating that in the existing experimental data and experimental environment, the recommendation performance of the proposed model is better than that of other recommendation models.

User-item CF is a traditional recommendation method that relies on the historical interaction data between users and items to calculate similarity. It is simple: modeling the relationships between users and items based on their historical ratings. However, it is sensitive to data sparsity and the long-tail effect, which can negatively impact its performance. Additionally, user-item CF struggles with the cold start problem, where it is difficult to recommend items to new users or recommend new items due to the lack of historical data.

Matrix factorization is a more advanced technique that decomposes the user-item rating matrix into two lower-dimensional matrices, capturing the latent features of users and items. Standard methods include singular value decomposition (SVD) and alternating least squares (ALS). This approach effectively handles large-scale datasets and often yields better recommendation results than traditional CF methods. However, it involves complex computations and parameter tuning and remains sensitive to data sparsity.

The latent factor model (LFM) is a specific implementation of matrix factorization that focuses on uncovering latent factors to explain user-item interactions. By mapping users and items into a low-dimensional space, LFM captures underlying preferences and

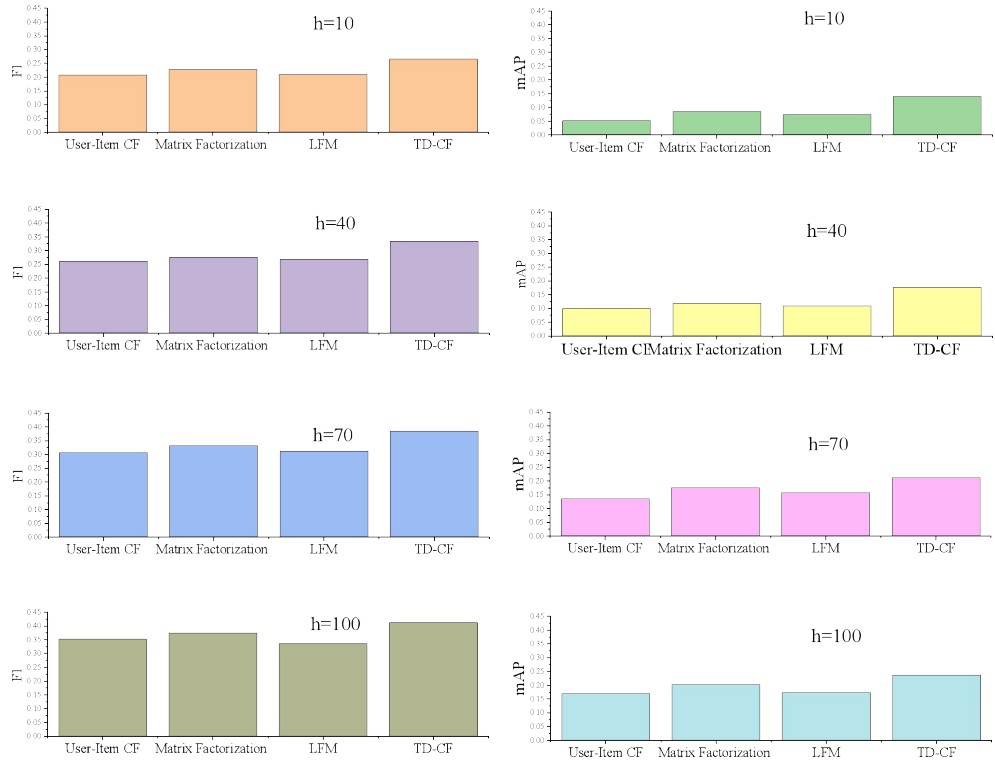

**Figure 4** **F1 and mAP of different models.**

characteristics. It provides good recommendation performance and is suitable for implicit feedback data. Despite these advantages, LFM requires careful parameter tuning and has a high computational complexity.

TD-CF introduces a time decay function into collaborative filtering, accounting for the temporal aspect of user interactions. TD-CF places greater weight on recent user behaviors by integrating the time decay function into similarity calculations and user preference estimation. This enhances the relevance of recommendations, especially for short-term user interests. The time decay mechanism helps mitigate the long-tail effect and data sparsity by reducing the influence of outdated interactions.

We show the computation time (seconds) and memory usage (MB) of each model in Fig. 5. User-item CF performs well on small-scale data, but with the increase of data size, the computation time and memory usage increase significantly, showing the characteristic of O(N 2). The computation time and memory usage of matrix factorization and LFM increase with data size growth; despite the high computational complexity, they can effectively handle large-scale data and mine latent features. After introducing the time decay function, the computational complexity of TD-CF is similar to that of user-item CF. Still, considering the time of User behavior improves the timeliness and relevance of

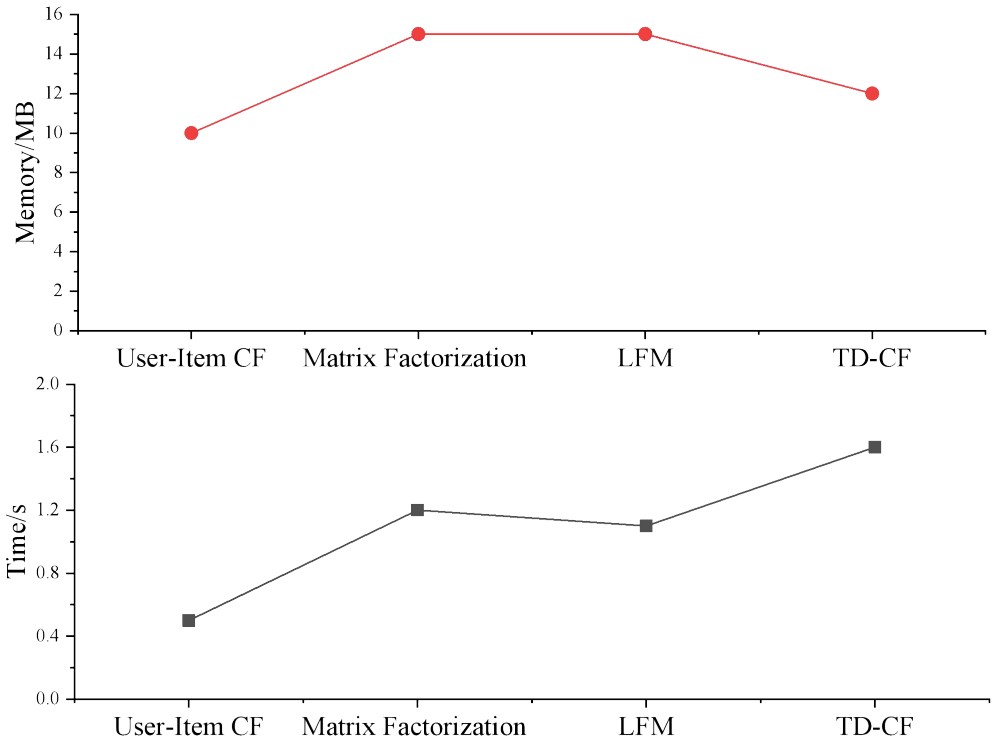

**Figure 5** Comparison of model complexity.

recommendation results. Although it increases some computational burden, it is still better than User-Item CF in performance.

The simulation results show that TD-CF maintains a low computational complexity while considering both recommendation accuracy and timeliness and is suitable for datasets of different sizes. Compared with other models, TD-CF has significant advantages in dealing with user dynamic interests and performs reasonably in consuming computing resources, which has high practical value.

The accuracy of the TD-CF model varies with weight $\alpha$, as shown in Fig. 6.

With the increase in the sample size, the model's accuracy is improved accordingly. When there are 40 and 50 songs, the accuracy of recommendation is the highest when $\alpha$ is 0.4; When there are 60 songs, the accuracy of recommendation is the highest when $\alpha$ is 0.5; When there are 70 songs, and $\alpha$ is 0.6, the accuracy of recommendation is the highest. This verifies the effectiveness of the mixed music recommendation model. When users have a rich listening history, the weight of the CF model is increased, and when users have less listening history, the weight of the TD model is increased. Thereby obtaining a better recommendation effect.

A linear decay provides a balanced weighting, ensuring that recent interactions are favored without overly diminishing the influence of slightly older preferences. Compared to exponential or logarithmic decay functions, the linear model is more accessible to interpret and control, avoiding scenarios where interest in older items diminishes too

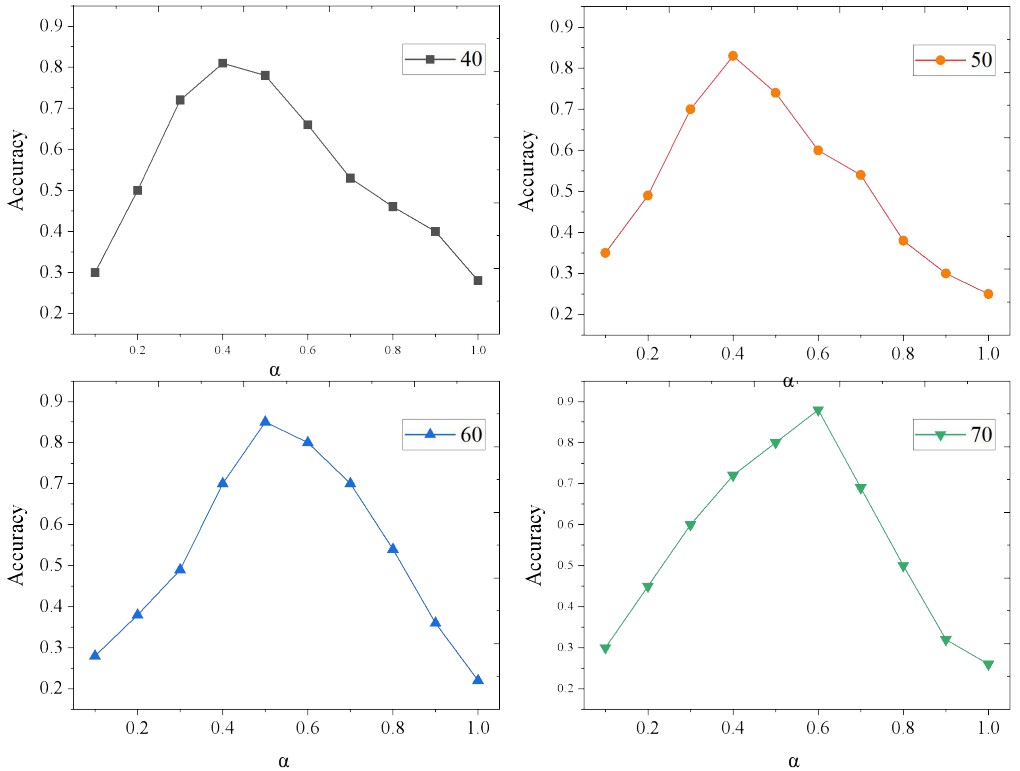

**Figure 6** Results of model accuracy changing with $\alpha$.

sharply (as with exponential decay) or too slowly (as with logarithmic decay). The linear function ensures a smooth and gradual reduction in the weight of past interactions, aligning well with the aim of maintaining a steady focus on recent user preferences in music teaching recommendations.

Figure 7 shows that for Last.fm Dataset-1k, the mixed recommendation algorithm in this paper, is more accurate under different recommended items. This is because the user's interest changes with time, and the user's forgetting of items is considered when calculating the similarity, thus improving the accuracy of predicting item scores. At the same time, it shows that the best recommendation effect can be obtained by selecting the appropriate number of recommended items $N = 50$.

In music teaching, TD-CF is selected to recommend teaching resources. By analogy with the characteristics of human memory and forgetting, the closer the time is, the more the user's interest can be reflected, and the higher the weight of short-term interest in recommendation is so that the characteristics of music teaching resources and users can be fully utilized to achieve more accurate recommendation.

## Discussion

The research tackles one of the primary limitations of traditional CF models: their sensitivity to data sparsity and cold start issues. By integrating the time decay function, TD-CF

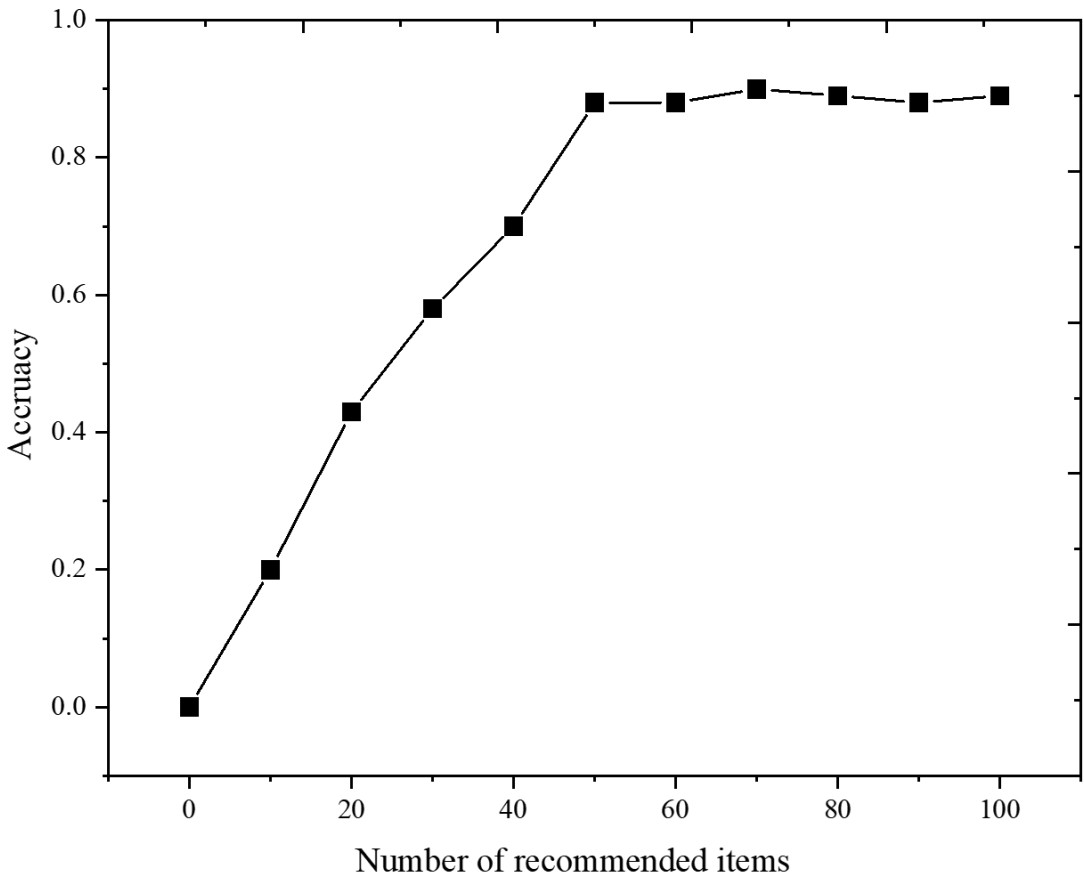

**Figure 7    Comparison of model accuracy under different number of recommended items.**

addresses these limitations by reducing the impact of outdated interactions and placing greater emphasis on recent user behaviors. This temporal aspect proves crucial in contexts like music recommendation, where user interests are highly dynamic and change over time. Figures 4 and 7 show that the TD-CF model consistently outperforms other models across different dataset sizes and recommendation list lengths, underscoring its robustness in handling short-term and long-term user preferences.

From a computational perspective, the results depicted in Fig. 5 highlight the efficiency of the TD-CF model. While matrix factorization techniques such as SVD++ and LFM exhibit superior performance in capturing latent features, they are computationally expensive and require complex parameter tuning. On the other hand, the TD-CF model maintains a low computational complexity (comparable to user-item CF) while improving recommendation timeliness and accuracy. This balance between performance and computational cost makes TD-CF a practical and scalable solution for large-scale recommendation systems, especially in educational environments where real-time recommendations are essential for improving learning outcomes.

Critically, our research advances the field by demonstrating how time-sensitive collaborative filtering can be applied to the domain of music teaching resources. In

this area, conventional recommendation systems fall short. Traditional models typically struggle to capture the evolving nature of user interests, which is particularly problematic in educational settings where student preferences and learning needs change over time. By leveraging time decay, the TD-CF model aligns more closely with the natural learning process, where more recent learning interactions are weighted more heavily than older ones. This analogy with human memory and forgetting proves instrumental in enhancing the recommendation of teaching resources, as it reflects the dynamic and temporal aspects of music consumption and education.

However, despite its effectiveness, the TD-CF model is not without limitations. One potential area for further research is the exploration of more sophisticated time decay functions that better capture the nuances of user behavior over extended periods. Additionally, while our model demonstrated high accuracy with an optimal value of $\alpha$, as shown in Fig. 6, further tuning of this parameter may be required when applying the model to different datasets or recommendation domains. Although mitigated by the time decay function, the model's reliance on historical data could still be improved by integrating additional contextual information, such as user demographics or content features, to refine recommendations further.

The proposed TD-CF model addresses fundamental limitations in traditional recommendation systems by introducing time decay. It is particularly well-suited for applications involving dynamic user preferences, such as music teaching.

## CONCLUSION

This research highlights the effectiveness of integrating time context information into traditional collaborative filtering (CF) algorithms, specifically through the proposed TD-CF recommendation algorithm. The verification results on the Last.fm Dataset-1k demonstrates that the TD-CF model's recommendation accuracy improves as the number of prediction scores increases, with a continuous decrease in the RMSE value. By incorporating time-sensitive data, the model effectively captures users' evolving short-term interests, leading to more precise recommendations. This is particularly useful in music education, where tailored recommendations can help optimize the use of music teaching resources by aligning them more closely with users' preferences.

However, despite the promising results, the study also has several limitations. Firstly, the dataset used in the study, while suitable for initial testing, may not fully represent the diversity of user behaviors and preferences in different music education environments. Future research should explore larger and more diverse datasets to ensure the model's robustness across various user groups and musical genres. Secondly, the study focuses primarily on short-term user interests. It does not delve deeply into long-term preferences, which could also play a significant role in improving recommendation accuracy over time. Incorporating a balanced approach that accounts for both short-term and long-term user behaviors could lead to more comprehensive recommendation systems.

Additionally, while the TD-CF model shows promising performance in music education, its applicability to other disciplines or multimedia recommendation systems remains

unexplored. Future studies could extend this model to different domains, such as video, literature, or other forms of educational content, to assess its versatility. Lastly, the model's reliance on time-based data may introduce biases toward more recent interactions, potentially overlooking valuable historical data. Future research could investigate hybrid models combining time-sensitive algorithms with content-based or knowledge-based approaches to address this challenge and improve recommendation performance.

## ACKNOWLEDGEMENTS

We thank the anonymous reviewers whose comments and suggestions helped to improve the manuscript.

### Funding
The authors received no funding for this work.

### Competing Interests
The authors declare that there are no conflicts of interest.

### Author Contributions
- Yina Zhao conceived and designed the experiments, analyzed the data, prepared figures and/or tables, and approved the final draft.
- Xiang Hua performed the experiments, performed the computation work, authored or reviewed drafts of the article, and approved the final draft.

### Data Availability
The code is available in the Supplemental File.

The raw data are available at Zenodo: Schifanella, R., Barrat, A., Cattuto, C., Markines, B., & Menczer, F. (2010). Last.fm Dataset [Data set]. Zenodo. https://doi.org/10.5281/zenodo.4589071.

### Supplemental Information
Supplemental information for this article can be found online at http://dx.doi.org/10.7717/peerj-cs.2533#supplemental-information.

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
