# Peer review of "Application of collaborative filtering algorithm based on time decay function in music teaching recommendation model"

_PeerJ Computer Science, doi:10.7717/peerj-cs.2533_

## Round 0.1 · original submission · Major Revisions

Please see the detailed reviews. The reviews emphasize the need for improvements across three key areas: basic reporting, experimental design, and validity of the findings. The paper demonstrates innovation in its recommendation system, particularly through similarity weighting and time decay functions. However, clearer articulation of the contributions in the abstract, introduction, and conclusion is necessary. The introduction should incorporate visuals, better problem formulation, and well-defined research questions. In the experimental design, distinct contribution areas in figures should be clarified, and performance metrics and comparisons with existing models should be organized. The reviewers suggest handling cold start problems, matrix sparsity, and more detailed performance comparisons. Including these enhancements will improve clarity, coherence, and the overall quality of the work.

Reviewer 1 ·

Basic reporting

1. Enrich the abstract by providing more detailed insights into the results achieved.
2. Consider enhancing the introduction by incorporating relevant visuals to aid in conveying key concepts.
3. Ensure the problem formulation in the introduction is thorough and clearly articulated.
4. Integrate well-defined research questions within the introduction to guide the reader through the study's focus.
5. Broaden the scope of the Related Work section by incorporating additional, relevant literature to strengthen the context.
6. Critically assess the existing literature and clearly highlight the key limitations, either in a well-structured paragraph or a comparative table.

Experimental design

7. In Figure 1, visually distinguish the specific areas where the authors have made their contributions.
8. Provide a comprehensive explanation of the evaluation metrics used in the study, ensuring clarity in their application and significance.
9. Organize the results in a tabular format and perform a detailed comparison with established standard techniques, highlighting the strengths and weaknesses.

Validity of the findings

10. Expand the conclusion to include a discussion of the study's limitations and suggest avenues for future research.
11. Incorporate a discussion section to thoroughly explore how your research addresses the identified problem and contributes to the field.

Reviewer 2 ·

Basic reporting

This method utilizes similarity weighting calculations to obtain a comprehensive representation of user emotions. While this paper exhibits elements of innovation, some modifications are necessary to enhance its overall quality.

1. The main contributions of the paper should be clearly highlighted in the abstract, introduction, and conclusion sections. As it stands, the manuscript does not adequately convey these contributions, making it difficult to assess the significance and impact of the research. A more explicit presentation of the study’s key findings and contributions is needed.

2. The construction of the user-music score matrix and the approach to generating recommendations based on user similarity are well-explained.

Experimental design

3. The description of constructing the user-music matrix and deriving recommendations is logically structured and aligns with standard collaborative filtering practices. The approach to handle missing ratings by considering the number of repetitions is a practical solution when explicit ratings are unavailable.

4. Handling Cold Start Problem: Provide strategies for addressing the cold start problem for new users or new music items that lack sufficient data.
5. Matrix Sparsity: Discuss how the model handles sparsity in the user-music matrix, as this is a common challenge in collaborative filtering.

6. Include a discussion on how the different similarity metrics (cosine and Jaccard) compare in terms of performance and accuracy in your specific context. Detail the process for tuning the weights ws and wt used in combining different similarity measures.

Validity of the findings

7. The use of a time decay function to adjust the weight of users' interests over time is a valuable addition, ensuring that recent user behavior is given appropriate consideration. But the author needs to discuss why a linear time decay function was chosen over other options like exponential or logarithmic decay. Include any empirical evidence or experiments conducted to justify this choice. Moreover, consider how the decay factor α\alphaα is adjusted dynamically based on user behavior or other contextual factors.

8. Performance Evaluation: Assess and compare the performance of the mixed model against the individual models to demonstrate the benefits of the hybrid approach.

Additional comments

9. While the reviewers acknowledge the importance of this work, particularly given the substantial patient sample size, the quality of the manuscript is compromised by frequent grammatical errors. These errors detract from the overall readability and professional presentation of the paper. A thorough proofreading and revision for grammatical accuracy are necessary to enhance clarity and coherence.

Reviewer 3 ·

Basic reporting

The proposed recommendation model is well-structured and incorporates several advanced techniques to enhance recommendation accuracy and relevance. The design is comprehensive, addressing key aspects of data processing, recommendation calculation, and prediction. However, further elaboration on performance evaluation, parameter tuning, and handling specific challenges like the cold start problem would strengthen the presentation of the model. Additionally, empirical validation and comparison of different strategies would provide a clearer picture of the model's effectiveness and robustness.

It would be helpful to include a brief roadmap or summary of the paper’s structure at the end of the introduction.

Provide a brief overview of how the TD-CF algorithm will address the identified problems in music recommendation. Include a summary of the key innovations and expected improvements over traditional methods.

Experimental design

Explain in more detail how the forgetting curve of human memory is used to fit the TD factor and why this approach is expected to be effective. This helps in understanding the theoretical foundation of the proposed algorithm.

Validity of the findings

The manuscript should include a comparison with other cognitive models used in similar studies. Specifically, it needs to outline how the novel method presented compares to these existing models and elucidate the specific benefits and improvements offered by the proposed approach.

Provide more details on how the weights for the CF model and the time decay model are determined. Is there a specific algorithm or criterion used to adjust

Additional comments

There are too many Chinese documents in the manuscript, and many of them are out of date. Cite specific studies or advancements that have attempted to address similar issues, and explain how your approach builds upon or diverges from these works.

---

## Round 0.2 · accepted · Accept

All reviewers have confirmed that the authors have addressed all of their comments.

Reviewer 1 ·

Basic reporting

All changes have been completed

Experimental design

All changes have been completed

Validity of the findings

All changes have been completed

Reviewer 2 ·

Basic reporting

'no comment'

Experimental design

'no comment'

Validity of the findings

'no comment'

Additional comments

'no comment'

Reviewer 3 ·

Basic reporting

All the issues reportes to the authors have been succesfully solved.

Experimental design

All the issues reportes to the authors have been succesfully solved.

Validity of the findings

All the issues reportes to the authors have been succesfully solved.